



# Identification of typical eco-hydrological behaviours using InSAR allows landscape-scale mapping of peatland condition

Andrew V. Bradley[1], Roxane Andersen[2], Chris Marshall[2], Andrew Sowter[3], David J. Large[4]

[1]Department of Chemical and Environmental Engineering, Faculty of Engineering, Nottingham Geospatial Institute, Innovation Park, Jubilee Campus, Nottingham, NG7 2TU, UK
[2]Environmental Research Institute, University of Highlands and Islands, Castle Street, Thurso, Scotland, KW14 7JD, UK
[3]Terra Motion Limited, Ingenuity Centre, Innovation Park, Jubilee Campus, University of Nottingham, Nottingham. NG7 2TU, UK
[4]Department of Chemical and Environmental Engineering, Faculty of Engineering, University of Nottingham, Nottingham. NG7 2RG, UK

*Correspondence to*: Andrew V. Bradley (andrew.bradley1@nottingham.ac.uk)

**Abstract.** Better tools for rapid and reliable assessment of global peatland extent and condition are urgently needed to support action to prevent their further decline. Peatland surface motion is a response to changes in the water and gas content of a peat body regulated by the ecology and hydrology of a peatland system. Surface motion is therefore a sensitive measure of ecohydrological condition but has traditionally been impossible to measure at the landscape scale. Here we examine the potential of surface motion metrics derived from InSAR satellite radar to map peatland condition in a blanket bog landscape. We show that the timing of maximum seasonal swelling of the peat is characterized by a bimodal distribution. The first maximum is typical of steeper topographic gradients, peatland margins, degraded peatland and more often associated with 'shrub'-dominated vegetation communities. The second maximum is typically associated with low topographic gradients often featuring pool systems, and Sphagnum dominated vegetation communities. Specific conditions associated with 'Sphagnum' and 'shrub' communities are also determined by the amplitude of swelling and average multiannual motion. Peatland restoration currently follows a re-wetting strategy, however our approach highlights that landscape setting appears to determine the optimal endpoint for restoration. Aligning expectation for restoration outcomes with landscape setting might optimise peatland stability and carbon storage. Importantly, deployment of this approach, based on surface motion dynamics, could support peatland mapping and management on a global scale.

## 1. Introduction

The conservation of functional peatlands and the restoration of degraded peatland, to reduce and ultimately mitigate land-use related emissions of atmospheric carbon dioxide, is now a global priority (Leifeld and Menichetti, 2018; Amelung et al., 2020; Günther et al., 2020). To support the implementation of national peatland management plans and restoration initiatives, cost-effective measures of current peatland condition and restoration progress are urgently required (Crump, 2017). Mapping peatland extent and condition has long been recognized as a huge challenge over large, remote, wet, and



often discontinuous peat forming regions where field-based surveys are impractical and expensive (Lees et al., 2018). Alternatives such as thematic mapping based on optical remote-sensing (visible and near-infra red) are increasingly used (Minasny et al., 2019), but the number of observations in regions with frequent cloud cover such as peatlands in the northern latitudes and the tropics reduces the number of possible surface observations. Radio detection and ranging (Radar) that is sensitive to physical properties of the surface, provides an effective, more frequent option, given that microwave frequencies can penetrate cloud cover and return a measured signal from the ground (Minasny et al., 2019; Poggio et al., 2014). For example, using the ESA Sentinel-1 Synthetic Aperture Radar (SAR) satellites it is now possible to observe a peatland surface anywhere at high frequency (6 to 12 days) with continuous spatial coverage. When this is combined with the technique of SAR Interferometry (InSAR) it allows detection of surface displacement, an indication of peatland condition, as a time-series of observations (Sowter et al., 2013).

In peatlands, the rise and fall of the surface, sometimes described as 'bog-breathing' (Kulczynski, 1949; Baden and Eggelsmann, 1964; Mustonon and Suena, 1971; Hutchinson, 1980; Kurimo, 1983; Almendinger et al., 1986; Price, 2003; Price and Schlotzhauer, 2003) is one of the key self-regulating feedback mechanisms providing resilience and maintaining function during periods of hydrological stress (Money and Wheeler, 1999; Waddington et al., 2015). This 'surface motion', which is a poro-elastic mechanical response to ecohydrological processes, results from the collapse and expansion of large pores in response to changes in the mass of water stored and associated stresses within the peat (Price, 2003). Mechanical deformation of the peat body and consequent surface motion can also modify the ecohydrology of a peatland via compaction, slope failure and pipe formation (Waddington et al., 2010; Waddington et al., 2015). Small-scale field observations indicate that peat surface motion is influenced by changes in water level (Roulet, 1991; Price, 2003; Kennedy and Price, 2005; Fritz at al., 2008; Alshammari et al., 2020), vegetation composition (Howie and Hebda, 2018; Alshammari et al., 2020), micro-topography (Waddington et al., 2010), accumulation and upward migration of methane bubbles (Glaser, et al., 2004; Reeve et al., 2013) and land management (Kennedy and Price, 2005).

Collectively these results suggest that peatland surface motion could be a sensitive indicator of peatland function on a landscape scale. So far, InSAR investigations have focused on discrete, small-scale (<1 km$^2$) peatlands (Fiaschi et al., 2019; Tampuu et al., 2020), identifying the potential range in timing and amplitude of seasonal peatland surface motion (Alshammari et al., 2020; Alshammari et al., 2018) and its relationship to precipitation (Fiaschi et al., 2019) water level (Alshammari et al., 2020; Tampuu et al., 2020) and vegetation composition (Alshammari et al., 2020). However, peatland landscapes contain a continuum of topographic, ecological, hydrological and management regimes and these small-scale studies have not captured the full spectrum of peatland conditions between degraded and near natural.

In this paper, we determine whether surface motion measured by InSAR can be used to quantify continuous changes in peatland condition over a complex peatland landscape. Using the APSIS (Advanced Pixel System using Intermittent SBAS) InSAR method formerly known as ISBAS (Sowter et al., 2013) (Materials and Methods) which is capable of generating spatially continuous measures of vertical surface motion over peatland (Sowter et al., 2013) we measure time series of



surface motion over our study site at a high spatial and temporal resolution. Specific time series metrics are then compared to independent measures of peatland condition to determine their relationship. By doing this we relate surface motion metrics to the continuum of ecohydrological conditions in this peatland landscape. Finally, we demonstrate how surface motion metrics can be used to map the ecohydrology of a peatland system. By doing so we illustrate how our new approach could be applied to monitoring the response of global peatlands to restoration, management, and climate change.

Our chosen study site is 930 km$^2$ of undulating blanket bog, ranging from 50 to 600 m.a.s.l in the Flow Country, Northern Scotland (Andersen et al., 2018; Fig. 1a). In the past, management has involved artificial drainage of the driest peatlands, targeted for subsidized agricultural improvement and later afforestation programs (Sloan et al., 2018). More recently, wetter near natural areas have been designated for conservation (Lindsay et al., 1988), and previously forested and drained areas are now undergoing restoration. Some areas are actively eroding, particularly at the highest altitudes (Hancock et al., 2018). This

complex mosaic of near-natural and modified peatland conditions within this study site makes it particularly suited to the use of InSAR mapping of peatland condition.









**Figure 1: Study location, Sub Sites (SS) and the surface motion metrics plots calculated from InSAR detected annual motion between May 10th 2016 to May 9th 2017. (a) The study location (inset) and study area, outlined, in the Flow Country, Northern Scotland. Forested areas are dark green, with the main river network shown. Plots for, (b) SS1 Balavreed, (c) SS2 Cross Lochs, (d) SS3 Knockfin Heights, (e) SS4 Loch Caluim, (f) SS5 Munsary. Axis: x, time (months); y, amplitude (amp.: m), z, velocity (vel.: m yr-1) with inset frequency (Freq.) histograms of peak timing in each of the sub sites. For velocity, +ve is upward and –ve is**

**downward motion of the surface. Histograms of peak timing at all sites display a bimodal distribution and the points on the scatter plots are colored to illustrate their relative position. Pale green and dark blue points are towards the front (low amplitude) and dark green and pale blue points are towards the back (high amplitude). Image sources for (a): ERSI World Imagery, sources Esri, DigitalGlobe, GeoEye, i-cubed, USDA FSA, USGS, AEX, Getmapping, Aerogrid, IGN, IGP, swisstopo, and the GIS User Community, © Crown copyright 2017. Distributed under the Open Government Licence (OGL). Ordnance Survey (Digimap**

**Licence).**

## 2 Materials and Methods

### 2.1 Data and time series processing

To calculate the surface motion, we used 410 Sentinel-1A and –1B synthetic aperture radar images (Descending Orbit 125) gathered every 6 to 12 days between 12/03/2015 and 01/06/2019 from the European Space Agency Copernicus Open Access

Hub (https://scihub.copernicus.eu). Satellite interferometry was applied using these images with the Advanced Pixel System using Intermittent SBAS (APSIS) technique. This technique, which was formerly known as the intermittent small baseline subset (ISBAS), is an advanced DInSAR technique (Sowter et al., 2013). The APSIS technique contains an adapted version of the established SBAS DInSAR time series algorithm (Bateson et al., 2015; Cigna and Sowter, 2017). It was designed to improve the density and spatial distribution of survey points to return measurements in vegetated areas, where DInSAR

processing algorithms habitually struggle due to incoherence (Osmanoğlu et al., 2016; Gong et al., 2016).

The APSIS algorithm was implemented using Terra Motion Limited's in-house Punnet software, which covers all aspects of processing from the co-registration of SLC (Single Look Complex) data to the generation of time series (54). Maximum horizontal baseline was restricted to 250m with maximum temporal separation of 1 year using a coherence threshold of 0.25 and point threshold of 360. Motion was measured relative to a stable reference point at Wick Airport (58.4533° N, 3.0879°

W). Phase unwrapping was implemented using an in-house implementation of the SNAPHU algorithm (55). Using APSIS, two products were produced for each georeferenced pixel location at approximately 80 by 90 m resolution. A motion time series of multiannual average line-of-sight velocity (m yr-1), and the time series of surface motion from which we are able to detect the seasonal expansion and contraction (or bog breathing) as annual oscillations in relative height (L. Alshammari, et al., 2018).

Each motion time series was processed as follows, to quantify the specific peatland surface motion metrics. Firstly, using the R programming environment (R Core Team, 2013), the time series was sub-sampled into equal time intervals of 12 days, to match the longest overpass interval of Sentinel-1 images since Sentinel 1B, which reduces overpass times to 6 days, was not operational until 2016. Outliers were re-estimated using the R 'tsclean' function (Box and Cox, 1964), from R package 'Forecast' (R. Hyndman et al., 2020). Gaps were filled with a linear interpolation using the R 'approx' function (Becker et

al., 1988) from R 'stats' package (R Core Team, 2020) after 'spline' methods were found to produce contradictory results



when considering adjacent time series across the largest gaps. The 'detrend' R function aligned and reset each time series around zero by subtracting the mean. Secondly, MSSA using the SSA-MTM toolkit (Ghil et al., 2002; SPECTRA, 2021) was applied to extract the cyclical seasonal trend from the 4-year 5-month time series. Covariance was calculated after channel reduction with Principal Component Analysis (PCA). Using a moving window of 12 months, long enough to capture

annual cycles, we calculated the first 10 PCA channels and 20 Empirical Orthogonal Functions (EOFs) to identify the seasonal cycles in the time series. In the first instance, surface motion time series were reconstructed using EOFs 1 - 6 (Fig. 2). This reconstruction captured the seasonal cycles but also included longer-term climate trends, notably three wetter years leading to the 2018 European wide drought (Buras et al., 2020). This climate trend causes merging and shouldering of peaks that compromised the detection of the seasonal cycles, particularly in the west of the study area, where it is wetter. To

overcome this difficulty, we used a surface motion time series reconstruction using EOFs 5 and 6 (Fig. 2; Supplement 1.1 Fig. S1) which extracted only the seasonal cycles. As the 2018 drought caused severe and widespread subsidence it subdued the multiannual average velocity. This was mitigated by recalculating multiannual average velocity for three complete motion cycles (March 2015 to March 2018) prior to the 2018 drought.



Earth **Surface**
**Dynamics**
Discussions

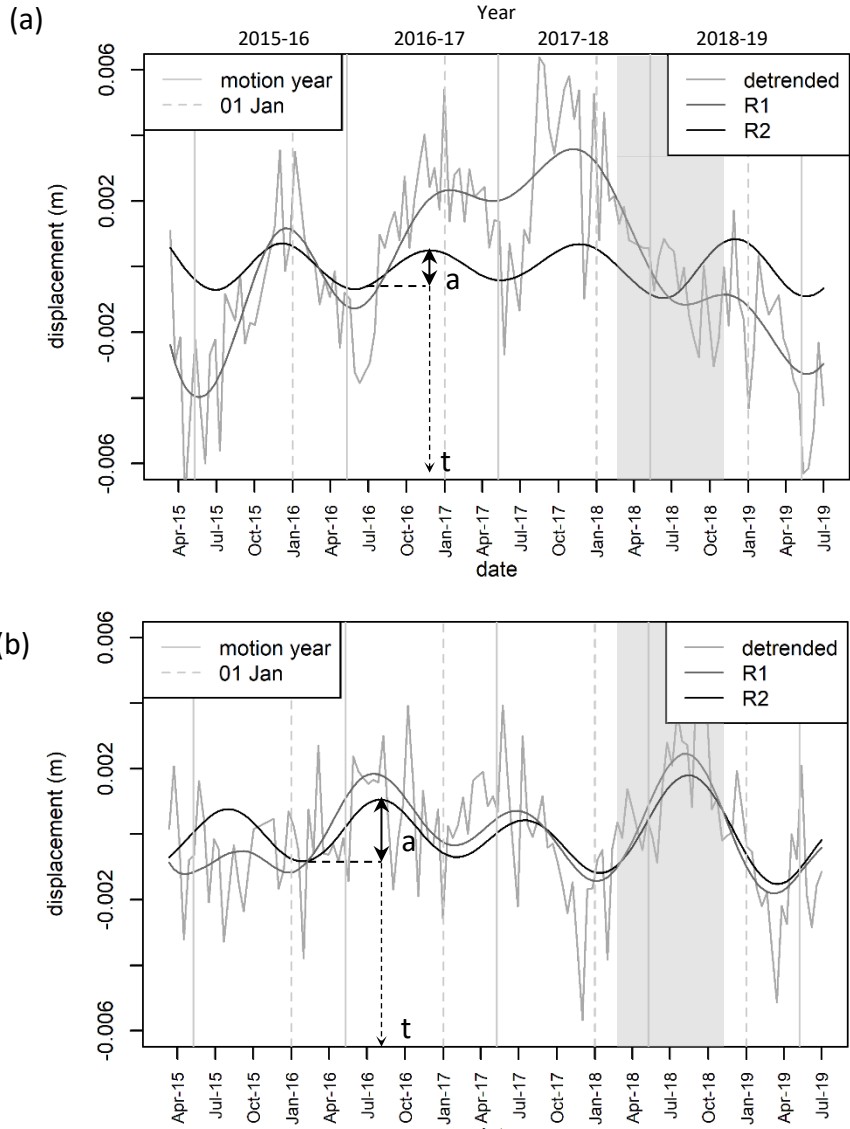

Figure 2: **Examples of surface motion time series and the metric definitions for (a) wet bog and (b) a drier bog, calculated from Sentinel-1 APSIS InSAR time series data between 12 March 2015 and 01 July 2019. The initial mean detrended time series (grey), and MSSA Reconstructions (R) retaining the local climate trend (R1, mid grey, a combination of empirical orthogonal functions 1 - 6) and annual seasonal cycles (R2, dark grey a combination of empirical orthogonal functions 5 and 6) are shown. The two surface motion metrics used in the analysis are, peak amplitude timing (dotted line, t), and amplitude (solid line, a) shown for the annual surface motion year May 10th 2016 to May 09th 2017. A third surface motion metric, multiannual average velocity is not defined here as it is part of the InSAR data processing (Materials and Methods). This asynchronous timing of peaks between (a) and (b) forms a bimodal distribution in the peak amplitude timing of the peatland landscape. The drought event of 2018 is indicated by the shaded column (Buras et al., 2020) and can be seen to influence the local climate trend in the wet bog (a). For MSSA details see Materials and Methods.**





Peak timing and amplitude of the seasonal cycles were extracted (Fig. 2) for individual years, using the R 'pracma' peak-find function (R Core Team, 2013). For measurement purposes, the start of the year was set to May 10th to avoid splitting the period in which the seasonal peak was likely to be detected. The measurement was not performed on the first and last years in the time series (2014 to 2015 and 2018 to 2019) as surface motion cycles are truncated preventing the accurate calculation of amplitude and peak timing. Surface motion time series were also tested to find multiple peaks per annum or years where

detection was not discernable, and these pixels were classed as having Irregular cycles (IRR). Irregular time series made up 8.4% of the data set and are commonly associated with water courses and damaged bog (including agriculture and forested areas) (Fig. 4). Exclusion of these irregular time series from the surface metrics plot (Fig. 1) does not affect our conclusions.

## 2.2 Ecohydrology of study area and sub-sites

The Flow Country peatlands exist in a range of topographic, hydrological and management settings, leading to a range of

different conditions e.g., from highly eroded to relatively intact peatlands, superimposed by activities such as forestry, drainage, and grazing across an undulating landscape. We predicted that the values of the three motion properties would spatially vary from one place to another showing as small variations on the shape of the 3-axis cluster plot as the values in in amplitude, velocity, and timing subtly shift from place to place. To demonstrate this, the five areas of peatland covering a range of conditions, roughly 10-15 km$^2$, defined as the sub-sites, were chosen based on local expert field knowledge and are

summarized in Table 1.

**Table 1: Details of the five subsites (SS), which are all currently designated as Site of Special Scientific Interest (SSSI), Special Protection Areas (SPA) and Special Areas of Conservation (SAC).**





| SS | 1 | 2 | 3 | 4 | 5 |
|---|---|---|---|---|---|
| Name | Balavreed | Cross Lochs | Knockfin | Loch Caluim | Munsary |
| Location | 58.38N -3.50E | 58.39N -3.94E | 58.32N -3.80E | 58.44N -3.68E | 58.39N -3.35E |
| Altitude (m.a.s.l) | ~180 | ~180 | ~360 | ~120 | ~100 |
| Topography | Watershed, gently undulating with pool systems | Flat pool systems on watershed with steep slopes into a valley | Eroding upland ridge with pools, ephemeral pools, and hags (wind eroded peat islands), | Gently sloping basin into central loch | Gently undulating area incised by small steams |
| General condition | Near-natural | Near- natural, drier peat | Eroding peat | Near- natural | Near natural surrounded by agricultural conversion and forestry |
| Current management | Low level sheep and deer grazing, conservation management agreement | Low to medium grazing by deer. Includes restoration (forest-to-bog and drain blocking) areas. Conservation management as part of Forsinard | Deer grazing, Forestry to the north and drainage to the East. Conservation management as part of FFNNR | Low level sheep and deer grazing, under conservation management agreement with FFNNR | Intense drainage for agriculture and grazing surrounded by forestry and forestry to bog to east and South. Part of the site under conservation management by Plantlife Scotland. |



| | | | | | |
|---|---|---|---|---|---|
| | Flows National Nature Reserve (FFNNR) | | | | |
| **History** | Evidence of damage from historic burning and drainage (hill drains) in places alongside natural drainage lines. | Surrounded by restoration areas (forest-to-bog undertaken in 2006) and standing forestry on deep peat. Wildfire in 1981 | The surrounding areas have been drained and burnt in the past | Some historic drainage and peat cutting. The area was also historically used for cattle grazing and is part of an old drove road. | Historic drain blocked with plastic piling, Historic drainage and burning |

### 2.3 Ecohydrological classification of the sub-sites

To identify the links between surface motion and the ecohydrology, the training bed of the sub-sites SS1 to SS5 were divided manually using Google Earth images, into 130 smaller polygons (most between $0.3 - 0.6$ km$^2$). To construct the polygons, one of the authors without specialist peatland knowledge used distinct contrasts where there were changes in the landscape structure, e.g. pool systems, visible drainage, vegetation reflectance and, evidence of land use management, e.g. fields and drainage as well as consistency in peak timing. In addition to the sub-site polygons, 125 random points were selected across the whole study area using the ESRI ArcMap "random point" tool. The immediate area close to the point was assessed for features in the landscape, (e.g., topographic setting, natural drainage, evidence of drainage ditches and where these features would influence hydrology, forests, restoration, management and the likely range, consistency or inconsistency in peak timing) and used to draw polygon boundaries to include this variation (most between $0.2 - 0.5$ km$^2$). While the sub-sites included the continuum of conditions and features adjacent to each other, the random polygons captured the ecohydrological state across the whole study area and avoided sample bias from the sub-sites. Measures of topography (average altitude, slope and aspect for all points in the polygon) were calculated from the Shuttle Radar Topography Mission Digital Elevation Model (Jarvis et al., 2008) for each polygon (Supplement 1.3).



The full set of polygons (sub-sites and random) was then passed to one of the authors with specialist peatland knowledge for a "blind" (i.e., without prior knowledge of or information about InSAR metrics) eco-hydrological classification. For each polygon, the cover of Plant Functional Types (PFTs); Sphagnum, other mosses, shrub, sedges, grasses, rushes, and conifer trees) and the presence of hydrological features (pools, streams, drains, erosion gullies, slope), were recorded using a semi-quantitative scale, (0 = not present or scarce, 1 = present, 2 = co-dominant, 3 = dominant). Current management (conservation, drainage for agriculture and peat cutting, forestry, restoration by forest-to-bog, and restoration by drain blocking) and historical management (e.g., burning, land-use conversion including wind-farm construction, restoration), was also documented for each polygon. This was achieved through a combination of existing data, field visits, local knowledge, 1:50 000 UK Ordnance Survey maps, NatureScot National Vegetation Classification maps (64), and Google Earth imagery. The author responsible for classification visited and surveyed all the sub-site polygons and 86 of the random polygons (85% of the polygons).

Using the semi quantitative scores, the vegetation and hydrology polygon attributes were clustered by similarity using a Hierarchical Cluster Analysis (HCA; Supplement 1.4, Fig. S3) to identify similar combinations of vegetation. To avoid an overly split hierarchical tree with only one or two members per cluster requiring complex explanation, it was deemed more informative to analyze the vegetation, hydrology and the topography category's separate from each other. For the vegetation, once the classes had been clustered, the average score for each category in the cluster was ranked with the top three PFTs used to characterize the plant functional group composition. Absence of a PFT was also noted (Tables S1-S4). For data visualization, clusters were grouped based on the dominant PFT, resulting in five groups: Sphagnum, Shrub, Grass, Bare peat (where Low or Absent vegetation was dominant) and Forestry. Very few clusters had were dominated by rushes (R) and all those had shrub as a co-dominant vegetation, so they were incorporated into the Shrub group. While sedges (Sg) were co-dominant in both Sphagnum and mostly shrub clusters, they were not the dominant PFT in any clusters and therefore did not form a separate group. These categories are used and expanded in the captions of Table 2.

**Table 2: Percentage proportion of clusters derived from Hierarchical Cluster Analysis based on plant functional types (PFTs) represented in the polygons of the five sub-sites and the random polygons. Clusters are defined by the dominant (first) and co-dominant (subsequent) PFTs. PFT notations: Sp = Sphagnum, S = Shrub, Sg = Sedges, M = Moss, G = Grasses, R= Rushes, F= Forest, LoA = Low or absent vegetation (Brash, bare peat following tree felling or restoration activities etc.). PFT in brackets denotes a notable presence. n=number of polygons. For data visualization, clusters were grouped based on the dominant PFTs in five groups: Sphagnum, Shrub, Grass, Bare peat and Forestry. Clusters dominated by Rushes (R) were incorporated in the shrub group for data visualization given their low number and shrub co-dominance. The group's Bare peat and Forestry were kept despite low numbers, as their vegetation is associated to specific management intervention.**

| Group | Sub-sites | SS1 | SS2 | SS3 | SS4 | SS5 | **Random** | All |
|-------|-----------|-----|-----|-----|-----|-----|------------|-----|





| Name | Clusters | | | % | | | Clusters | % |
|------|----------|------|------|------|------|------|----------|------|
| SPH | Sp,S,Sg | 37.9 | 29.6 | 0 | 45.5 | 19.2 | Sp,Sg,S | 28 |
| SHRUB | S,Sg,Sp(G) | 17.2 | 14.8 | 0 | 13.6 | 0 | S,Sg,M/ Sp | 28 |
| | S,Sg,R | 10.3 | 48.1 | 3.8 | 4.5 | 34.6 | S,Sg,G,M | 8 |
| | S,Sg,M | 20.7 | 0 | 69.2 | 9.1 | 0 | S,G,R | 8 |
| | S,Sg,M(G) | 3.4 | 0 | 23.1 | 18.2 | 0 | | |
| | R,S | 0 | 0 | 0 | 0 | 3.8 | R,Sg,S | 4 |
| GRASS | G,S,R | 10.3 | 3.7 | 0 | 4.5 | 11.5 | G,R,S, nSp,nM | 8 |
| | G,R | 0 | 0 | 0 | 4.5 | 11.5 | G,S,R | 1.6 |
| BARE | LoA | 0 | 0 | 0 | 0 | 7.7 | LoA | 4 |
| FOR | F | 0 | 3.7 | 3.8 | 0 | 3.8 | F | 9.6 |
| | *n=* | *26* | *27* | *26* | *22* | *29* | *n=* | *125* |

We also categorized topography into 1 = low, 2 = med and 3 = high, and split slope face direction into four quadrants and passed the data through an HCA. Except for the eroded SS3 site, altitude and aspect did not show any meaningful cluster groups and played no further part in the analysis. The lack of topographic relationships are largely due to the gentle relief of

the Flow Country that has few sheltered slopes and valleys. Instead, we used average gradient (degrees) in the polygon and found a natural breakpoint at 1.5 degrees that split the dataset equally between FLAT (< 1.5 degrees) and SLOPE (>1.5 degrees), with most pools with Sphagnum found in FLAT (Fig. 3a).

(a)



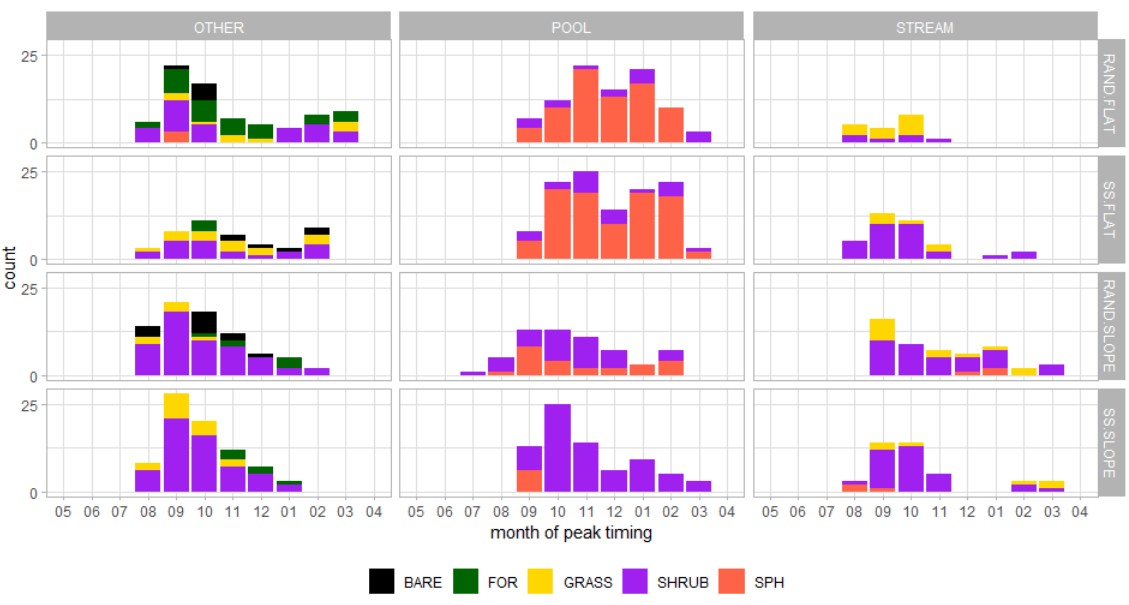


(b)

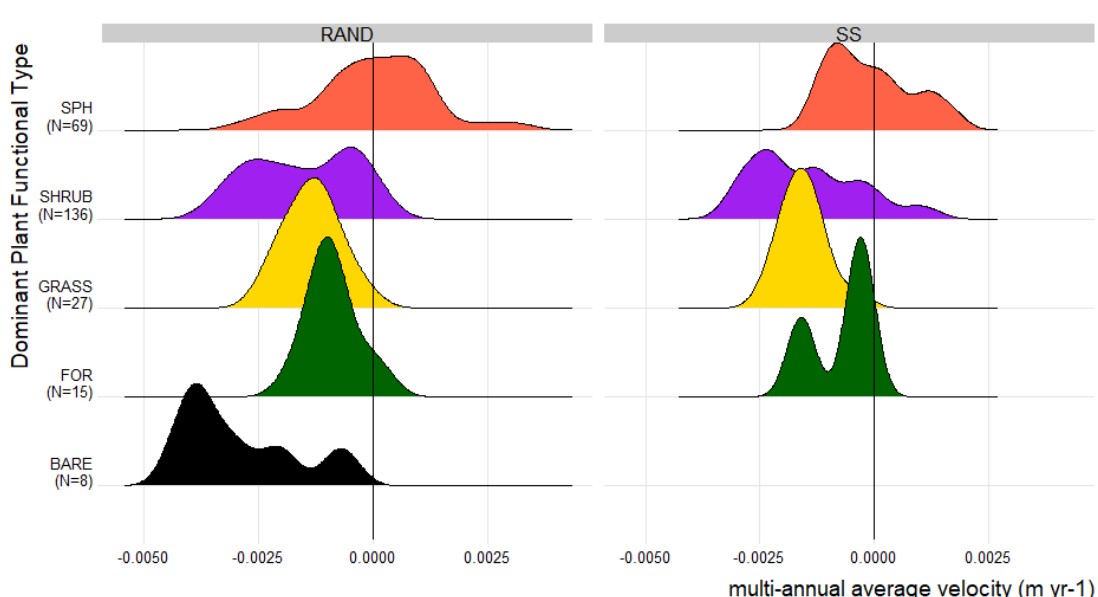

(c)





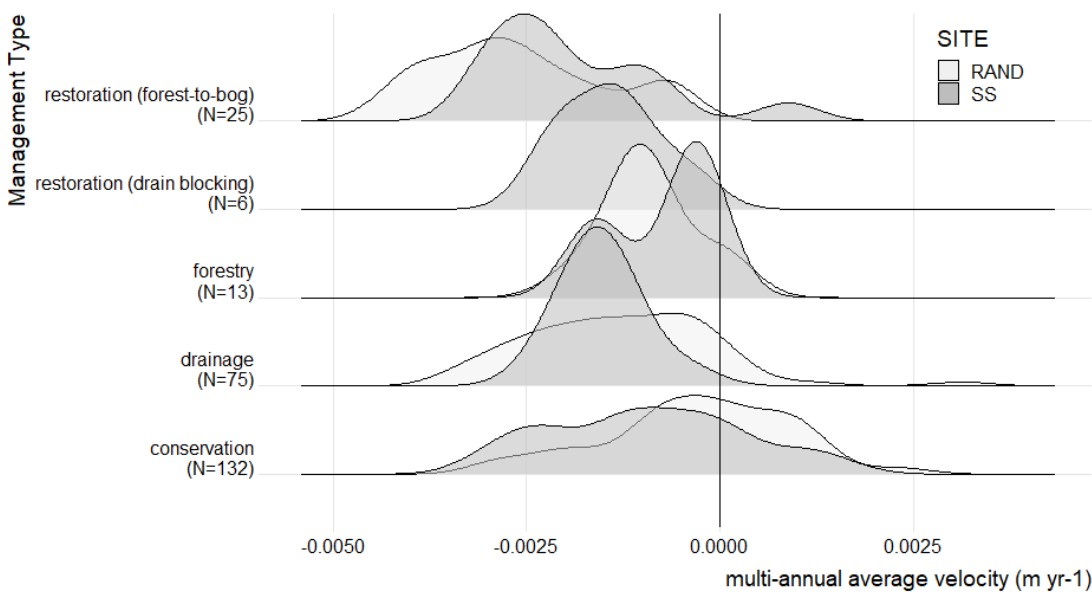

(d)

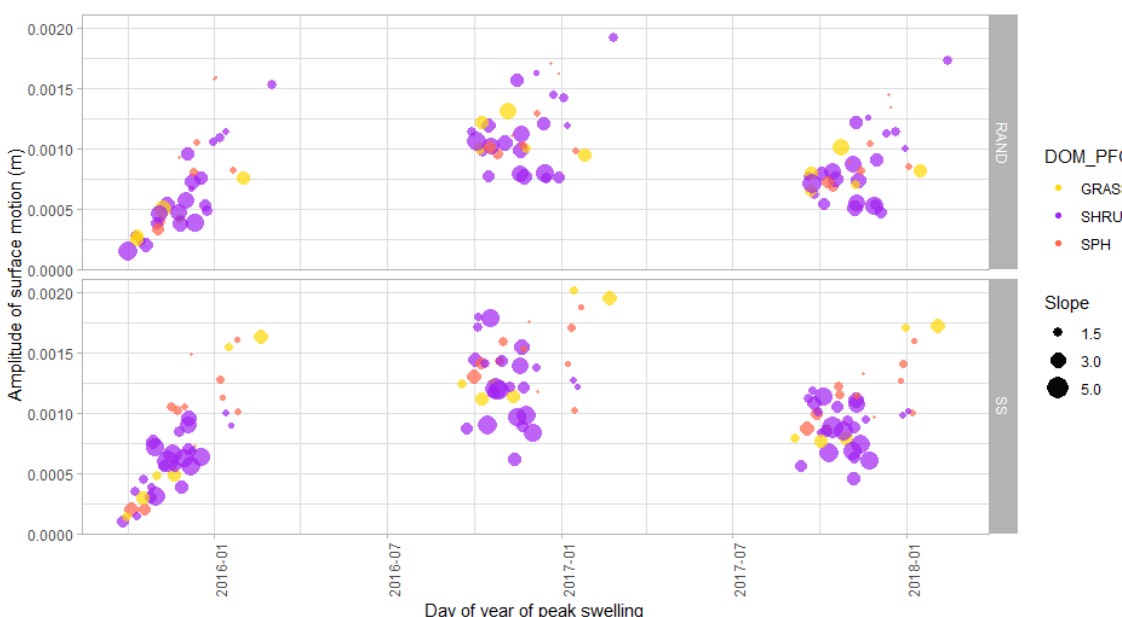

**Figure 3: Polygon summaries with respect to dominant plant functional clusters, management groups, topography and the three motion metrics. (a) Distribution of median polygon peak timing over time (month) for dominant plant functional clusters for polygons with pools, streams or other hydrological feature (e.g. drains, erosion gullies, peat cutting, or no apparent features) in**

**either FLAT (gradient < 1.5°) or SLOPE (gradient > 1.5°) topographic setting. Plant functional clusters: BARE = bare peat, FOR = conifer plantation, GRASS = grass-dominated communities, typically Molinia caerulae, SHRUB = shrub dominated communities, typically Calluna vulgaris and/or Erica tetralix, SPH = Sphagnum dominated communities. Sedges, rushes and other mosses are also present, often as co-dominant species in both SPH and SHRUB communities (see Table 2). Months are numbered**



from May (05) through to the following April (04). (b) Joy plots showing the variation in multiannual velocity for each plant
functional group, polygon type (RAND= Random, SS=sub-site) and topography. (c) Joy plots of multiannual average velocity for
different management groups (restoration, forestry, drainage, conservation) by polygon type (RAND, SS). (d) The timing of and
relative amplitude for three consecutive years (2015-2018) with respect to slope gradient (degrees), dominant plant functional
clusters (GRASS, SHRUB and SPH), and by polygon type (RAND, SS).

Summary statistics of the three surface motion metrics were made for each polygon. Although one side of the binomial

distribution usually dominated, the median score of each polygon was used over the mean. This is because the transitional

nature of the environment (i.e. a wet peat center has drier edges with slightly different vegetation composition), where

polygon edges may consist of some pixels from the opposite side of the bimodal distribution (Fig. 1b-f) causing a slight

skew in their distributions. The use of the median score was found to reduce these effects for Fig. 3d.

### 2.4 Mapping the state of the peatland system

To test whether the metrics of peak timing, amplitude and multiannual velocity can be used to map and predict peatland

condition we classified the position of pixels within the surface motion metrics plot. The classification was based on the

Euclidian distance from what restoration practitioners would consider a reference point corresponding to a good condition

'wet' Sphagnum peat. This is characterized by high amplitude (e.g., 0.008 m), a positive multiannual velocity (e.g., 0.006 m

yr-1), and the most frequent peak timing of the 'wet' Sphagnum dominated condition (e.g., February) within a plot of

amplitude vs. peak timing vs. multiannual velocity for the entire study area. The actual reference point was selected by

stepping down through the percentiles of the metrics distributions until the case with, the most frequent timing, highest

positive velocity and highest amplitude was identified. It would also be possible to integrate field observations and choose

the reference point values from specific pixel(s) if a particular condition was to be investigated. Data for each pixel were

paired with the reference point and the Euclidian distance in 3-dimensional (Cartesian) space was calculated. If the paired

pixel was in the 'dry' shrub side of bimodal distribution, the Euclidian distance was mapped as a 'V' shaped path via zero

velocity and zero amplitude at the mid date, 10th November, between the 'wet' Sphagnum and 'dry' shrub conditions. Prior

to calculation, the positions of the outer portions of the 'wet' and 'dry' distributions were adjusted. This is because if the

paired point is earlier than (left of) the dry peak and later than (right of) the wet peak the difference between the peak timing

and the origin would be incorrectly estimated. To mitigate this, these cases were folded inwards along the axis of the peak of

their distributions (effectively turning the upturned 'W' shape of the bimodal distribution into an 'M' shape). Using the

natural breaks (Jenks) classification as a guide in ESRI ArcGIS, thresholds were used to map 'wet' Sphagnum dominated,

'dry' shrub dominated and the thin/modified peat classes across the whole study site to produce an ecohydrological map

(Fig. 4d).

To verify the predictive accuracy of the ecohydrological map we remotely identified the central locations of all the pool

systems (328 in total), within the study area using Google Earth images, and compared if these points corresponded to the

Sphagnum 'wet' condition. To capture the wider complex morphology, varying geometries (and sometimes variable

condition) of a full pool system (Lindsay, 2016), a search area of 150m (using the buffer function in ESRI ArcMap) was then





calculated. This buffer area contained at least 3 by 3 pixels of the ecohydrological map. We than calculated the percentage of Sphagnum 'wet' pixels in the buffer. Pixels classed as irregular were not included in the count (Table S5).





(a)

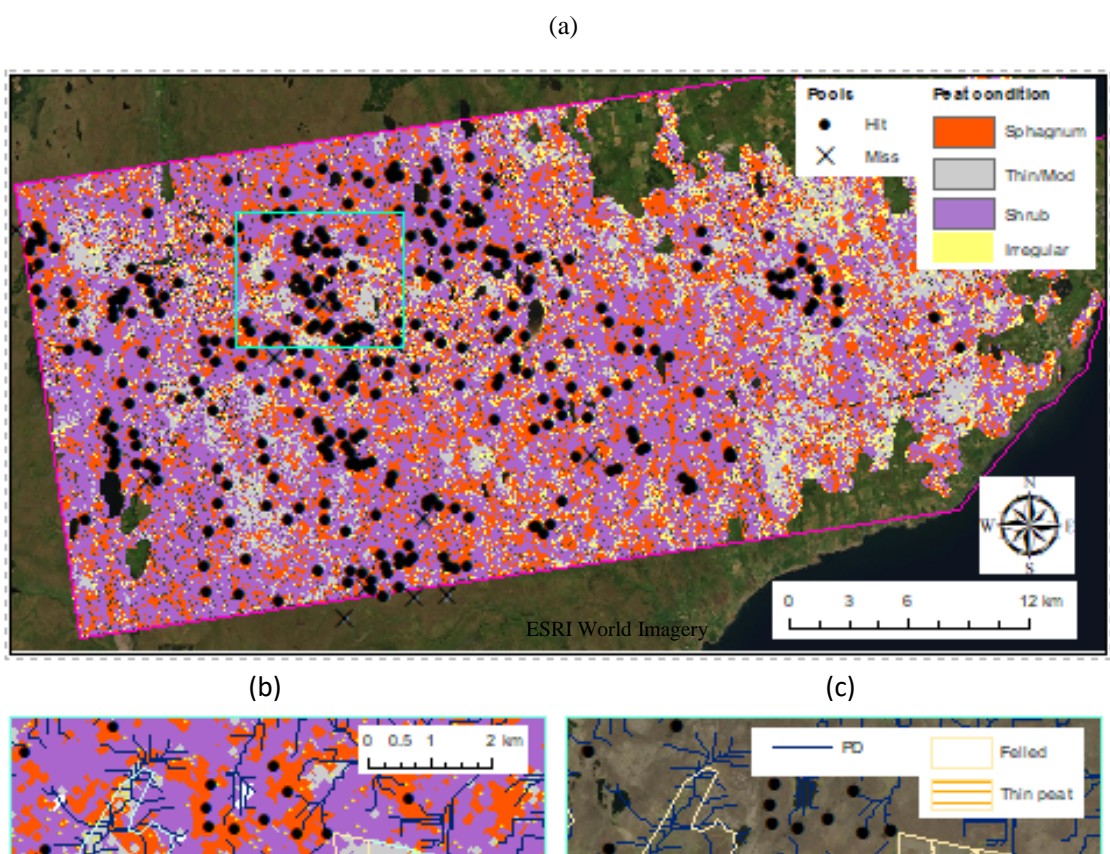

(b)                                    (c)

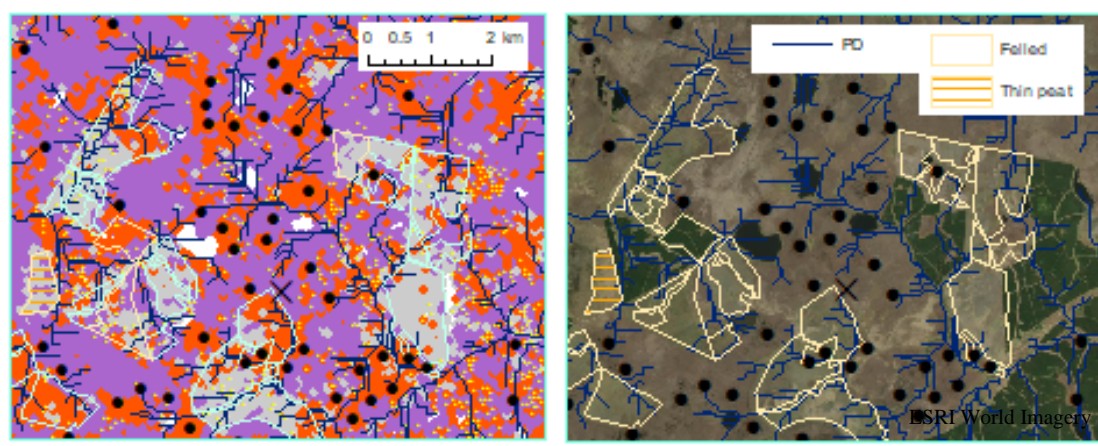

(d)

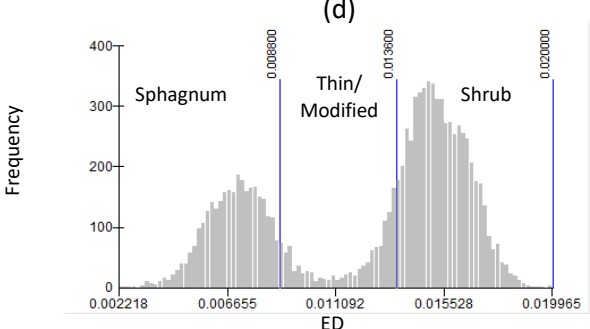




**Figure 4: The classification of peatland condition, with respect to the location of pool systems, and areas of forest-to-bog restoration in the study area. (a) Classified map based on the Euclidian distance (in 3-dimensional Cartesian space) from the position of a 'wet' Sphagnum endmember of each point in the plot of peak timing, amplitude and multiannual velocity for the period June 2016 to May 2017. Three peat classifications are indicated, Sphagnum, Thin/modified, and Shrub. Pixels that could**
**not be classified due to an absence of distinct seasonal oscillation are classified as irregular. The classified area (approx. 930 km$^2$) was delineated using peat soils from the National Soil Map of Scotland (JHI, 2021). Points in pool systems that have been correctly classified are shown as a "hit" and those incorrectly classified as a "miss". (b) A detailed view of the classified area highlighted within (a) illustrating the relationship with potential drainage (PD), determined from a DEM, areas of thin peat (hatched), and peat areas at various stages of forest to bog restoration (outlined) ranging from recently felled (thin/modified class) to almost fully**
**restored via rewetting (Sphagnum class). (c) A true colour satellite image of area, (b). Some of these restoration areas now have conditions of wet peat. (d) Frequency distribution of Euclidian distance, and the thresholds used to highlight the three peat conditions, Sphagnum dominated, Thin/modified and Shrub dominated. Images sourced via ESRI ArcMap in 2021. The variability seen within the felled forest blocks reflects variable degree of recovery associated with varying starting conditions, time since initiation of restoration (ranging from 0 to >15 years), landscape position and technique of the intervention. Image source for**
**(a) and (c) ESRI World Imagery: Esri, DigitalGlobe, GeoEye, i-cubed, USDA FSA, USGS, AEX, Getmapping, Aerogrid, IGN, IGP, swisstopo, and the GIS User Community.**

## 3 Results

### 3.1 Surface motion in blanket peatland

InSAR surface motion time series and long-term average motion were generated for the period March 12th 2015 to July 1st
2019. In the timeseries, the height of the surface was calculated relative to its first point and was determined every 6 days at pixel resolution 90 x 80 m across the study site. Seasonal cycles, subsiding during the summer and rising in the winter were observed in time series from the majority of pixels. From this, we defined a motion year beginning at the least active period in May and thus avoided dissection of the peaks in the distribution. We then focused our analysis on the period May 10th 2015 to May 9th 2018 to minimize uncertainties from a drought in mid to late 2018 (Buras et al., 2020), and illustrate data
for the central year, 2016 to 2017, where the buildup and continuation of adjacent years is complete.

Multichannel Singular Spectrum Analysis (MSSA) was applied to the dataset to isolate this cyclical, annual seasonal component of the time series (Fig. 2) and the following three surface motion metrics used to represent the condition of the peatland within each pixel. Metric one, the timing (date) of the annual peak in the seasonal cycle within 12 months from mid-May (Fig. 2). This has been shown to relate to peatland ecohydrology (L. Alshammari, et al., 2020; Tampuu et al.,,
2020). Metric two, the annual maximum amplitude (m) in surface motion measured from the previous seasonal minimum (Fig. 2). This is an indicator of the elastic response of the peat to changes in water storage (Roulet, 1991; Waddington et al., 2010). Metric three, the multiannual average vertical velocity (m yr-1) of the peatland surface. This is a measure of peatland growth (positive value) or subsidence (negative value) calculated over a fixed section of the time series (Sowter et al., 2013; Materials and Methods).

To understand the variation and distribution of our metrics across a spectrum of peatland conditions, we plotted them over five well-documented 10 to 15 km$^2$ sub-sites within the study area. These sub-sites span a range of landscape settings and ecohydrological conditions, from a gently undulating low-lying watershed with well-developed pool systems to actively eroding high plateau (Table 1, Fig. 1a). Plots of the metrics from the sub-sites all show features of a bimodal data





distribution with respect to annual peak timing (Fig. 1b-f). The bimodal distribution peaks fall between August to October

and December to February. Each sub-site shows a variation in the range of amplitude and velocity (Fig. 1b-f), reflecting the

diversity of peatland conditions sampled across the landscape.

**3.2 Relationship between surface motion and eco-hydrology**

To understand the link between the surface motion metrics and peatland condition, we divided the sub-sites into 130

polygons (0.2 to 0.6 km$^2$) in which metrics were aggregated. Additionally, 125 similarly sized individual polygons were

generated, randomly distributed across the whole study area. This size was practical for reliable field and map-based

validation and meaningful for capturing key features of the landscape (e.g., pool systems, forestry or restoration block,

stream and banks). Dominant plant functional types (Sphagnum, sedges, shrub, grass, bare peat, forestry), hydrological

features (pools, erosion gullies, streams, drainage ditches), topographic setting (slope, elevation) was recorded for each

polygon and used for eco-hydrological classification (Materials and Methods; SI Appendix).

A hierarchical clustering approach (HCA; Fig. S3) revealed ecological groups relating to dominant plant functional types

comparable between sub-sites and random polygons (Table 2) as well as hydrological groups separating polygons with pool

systems and those with streams from all other polygons. Comparison of the HCA based classifications and topographic

information (slope) to the surface motion metrics enables consistent relationships to be determined for sub-sites and random

sites, as follows. First, shifts in the polygon monthly peak timing distributions relate to a combination of topography,

hydrology, and plant functional group (Fig. 3a). Within the hydrological class POOL, polygons with topographic gradients

greater than 1.5°(RAND SLOPE, SS SLOPE), have their highest monthly frequencies earlier in September and or October

than polygons on flatter ground with topographic gradients less than 1.5°(FLAT), which tend to be November, January and

February. The steeper gradients, RAND SLOPE, SS SLOPE, tend to be associated with SHRUB and GRASS dominated

vegetation types with low frequencies of Sphagnum dominated polygons (SPH). Polygons with POOL as the dominant

hydrological feature tend to have their highest frequencies from October onwards, later than polygons with STREAM or

OTHER, independent of being FLAT or SLOPE. Sphagnum-dominated polygons (SPH) are almost exclusively associated

with POOLS and flat ground (RAND FLAT, SS FLAT), and tend to have their highest monthly frequencies later in the year

(November to February) than the other hydrological and topographical situations which are mainly SHRUB dominated

polygons as well as the other PFTs, such as GRASS, FOR and BARE.

Second, the most positive values of multiannual average velocity were almost entirely dominated by Sphagnum (SPH, Fig.

3b). Polygons with plant functional types typically associated with natural or man-made drainage (SHRUB), disturbance

(forestry (FOR) and bare peat (BARE) or thin, degraded peat (GRASS) consistently displayed negative long-term

(multiannual) average velocities regardless of topographical setting. Sites in which grass or forestry dominate tend to have a

more intermediate multiannual average velocity than either SHRUB or Sphagnum (SPH) dominant polygons. Where bare

peat is dominant, the most negative velocities occur.





Third, when multiannual average velocities are compared across different management classes (Fig. 3c), the least negative values are observed under conservation management and most negative values are associated with forest-to-bog management, a restoration approach that typically involves compaction from heavy machinery during the removal of conifer stands, followed by drain blocking and surface re-profiling. This restoration class shows a broader distribution in long term

multiannual average velocity than other management classes, reflecting variable degree of recovery associated with differing starting condition, time since initiation (ranging from 0 to >15 years) and techniques used in the intervention.

The factors controlling amplitude can be deduced from amplitude and peak timing plots for the three most dominant PFT clusters (SHRUB, SPH and GRASS) across three surface motion years (Fig. 3d). These graphs all show a strong linear relationship between timing (day of year) and amplitude with higher amplitudes occurring later in each surface motion year.

Steeper slopes are more likely to have lower amplitudes that peak earlier in each surface motion year. Shallower slopes are more likely to have higher amplitudes and peak later in the surface motion year. There is also year-on-year variation in seasonal amplitude, likely to be related to interannual variation in the local water table coupled with the amount of available unfilled pore space in the uppermost layer of the peat. In this sequence the polygon sites indicate that the peatland became gradually more saturated, to the point that by 2017-2018 there were relatively fewer pores to fill, and annual surface motion

was reduced.

Synthesizing the above, the bimodal distribution of peatland surface motion timing within our landscape may be interpreted as reflecting two dominant components of the landscape, a drier shrub dominated and a wetter Sphagnum dominated component. Wetter, flatter sites in a 'near natural condition', typically dominated by SPH PFT tend to reach peak surface heights later in the year, have higher amplitudes and stable-positive velocities. Drier SHRUB and GRASS PFT dominated

sites tend to reach peak surface heights earlier in the year, have lower amplitude oscillations and negative velocities. That the distribution of surface motion metrics in this particular blanket bog landscape is bimodal reflects a combination of the natural state of the peatland and the legacy of past management.

### 3.3 Application to large area condition mapping

The observed relationship between surface motion metrics and ecohydrology is readily interpreted in the context of reported

field measurements of peat surface motion (Howie and Hebda, 2018; Morton and Heinemeyer, 2019). Flatter sites under near natural conditions are poorly drained, wetter and dominated by Sphagnum spp. In turn, Sphagnum spp have a considerable capacity for water storage as a direct result of their physiology (Kellner and Halldin, 2002), resulting in peak water storage and seasonal swelling of the surface late in the year. Drier sites with compacted peat have less capacity to store water and reach water holding capacity earlier in the autumn (Price, 2003). Furthermore, the more degraded peat in these

sites is less elastic and therefore exhibits a lower amplitude response to changes in water storage (Holden et al., 2004; Lui and Lennartz, 2019). As the seasonal water balance shifts, drier, better drained sites lose water first followed by the



Sphagnum sites which may continue to swell on account of a hysteresis during the first stages of water loss (Howie and Hebda, 2018).

With this interpretation of the relationship between the surface motion metrics and ecohydrology, it should be possible to use the InSAR time series to map peatland condition. To illustrate this approach and evaluate its potential, a classified condition map was generated (Fig. 4a). This map was based on the classified Euclidian distance (Materials and Methods) from an ideal 'wet' Sphagnum-dominated reference point (positive velocity, high amplitude, late winter peak timing) within a plot of amplitude vs. peak timing vs. multiannual velocity for the entire study area. The resulting histogram of Euclidian distance (Materials and Methods) was split into three broad peatland classes (Fig. 4). A Sphagnum class characterized by winter (December to February) peak timing, high amplitude, and stable to positive velocity). A shrub class characterized by distinct autumn (August to October) peak timing with lower amplitude and negative velocities. A thin/modified peat class, characterized by both low amplitudes, negative velocities, and peak timing dissimilar to either the 'Sphagnum' or 'shrub' class. These thin/modified areas are expected to correspond with the most degraded and drained grass dominated sites or sites under restoration that are in transition to either a Sphagnum or shrub dominated state.

To assess the predictive accuracy of the Sphagnum class we determined the proportion of 328 pool systems that coincided with pixels of that class. Although wet areas in which Sphagnum is dominant do not necessarily contain pool systems, the reverse is nearly always true, and in the study area pool systems provide a spatially distributed, abundant and easily identifiable sample of this part of the peatland system. They also correspond to the part of peatland systems most unequivocally associated with 'near-natural' ecohydrological condition. The shrub and thin/modified classes are more likely to correspond with areas in between on sloping, degraded, forested and formerly forested peatland, some of which may have been much wetter prior to drainage.

A single marker was positioned in each selected pool system and the accuracy of the method determined based on whether this marker lies within 150 m of a correctly classified pixel. There is a need to tolerate a level of uncertainty as pool systems often display complex geometry related to local hydrology (Goode, 1973; Lindsay, 2016) and the position of the marker point could not take this into account. On this basis, identification of wet peat conditions around the pool systems is 97.9% accurate. Detailed inspection of the remaining 2.1% (7/328) of pool systems that did not fall within our threshold reveals that these pool systems all showed evidence of localized erosion or drainage causing degradation of their natural hydrology. From this, we can deduce that our method is converging towards 100% accuracy in identifying Sphagnum dominated pool systems in a near natural ecohydrological condition.

Inspection of our classification relative to other known features indicates that the thin/modified class corresponds to areas under restoration, notably areas recently felled for forest to bog restoration, areas subject to intensive grazing, thin peat soils on steeper higher ground and in valley bottoms (Fig. 4b,c). The abundance of the thin/ modified class is striking in the east of the study area that corresponds to long-term historical usage of the land for agriculture and associated cutting of peat for fuel (Andersen et al., 2018; Minasny et al., 2019).



Our classification also provides an overall measure of the state of this blanket bog landscape, to which future regional change, on account of climate change or restoration, may be compared. For example, within the area, our method identifies approximately 254 km$^2$ (27.3 %) of the area) as wet Sphagnum dominated peat, 481 km$^2$ (51.7 %) as shrub dominated peat, 117 km$^2$ (12.8 %) as the thin/modified peat class with 78 km$^2$ (8.4 %) as irregular time series.

## 4 Discussion

Our most important finding is that surface motion metrics derived from APSIS InSAR time series enable almost continuous spatial and temporal characterization of peatland condition at large scales. That the SAR data can penetrate cloud cover, measures regular physical displacement of the surface, and captures a known dynamic behavior associated with peat resilience gives this approach a significant lead over the far more challenging effort to measure peatland condition from optical reflectance data. This is compounded by the fact that cool wet peatlands are often obscured by cloud (Minasny et al.,
410 2019).

The sensitivity and dynamic response of surface motion metrics to changes in the state of the peatland system should make the method ideally suited to monitoring and informing peatland management and restoration. Globally, large areas of northern peatland degraded by historic drainage, grazing and forestry are now under or targeted for restoration (Rochefort et al., 2017). As a consequence, peatland conservation and restoration are increasingly perceived as critical tools in the fight
against global climate change (Leifeld and Menichetti, 2018; Amelung et al., 2020; Günther et al., 2020). Restoration strategies typically involve raising water levels to re-establish wet conditions. The expectation is that this will promote Sphagnum establishment, often a key measure of the success of an intervention (Rochefort et al., 2017; Bellamy et al., 2012; Caporn et al., 2018; González and Rochefort, 2019).

In the case of blanket bog landscapes, our finding of naturally drier shrub and wetter Sphagnum states raises the question as
to whether a linear strategy of increasing peat wetness is always an appropriate restoration target, or indeed if it is the only desirable outcome in all peatland settings. In this context, an APSIS InSAR-based assessment of the condition of a whole peatland can help guide restoration strategies by, firstly identifying the typical natural states and hydrological structure of that peatland, and secondly, following intervention, this approach could enable a robust monitoring of restoration trajectories and outcomes.

In natural landscapes, these peatland states are a consequence of landscape evolution in which the vertical accumulation of peat must be counterbalanced on an appropriate spatial and temporal scale by erosion (Large et al., 2021). Drier states correspond to areas of net carbon loss due to natural drainage, incision and erosion along peatland margins, and wetter states correspond to peatland interiors, areas with low gradient, that tend towards carbon accumulation. In this context to restore a site that is naturally dry to the wet state would risk instability, while the opposite would fail to optimize carbon storage. A
more suitable and sustainable ambition is to accept that restored blanket bog sites may follow different trajectories towards naturally Sphagnum or shrub states, and that these target end states will be constrained by the hydrological landscape setting,

as conceptualized by Winter (Winter, 1988). Our approach provides evidence for these natural states co-existing within the study areas, and evidence to guide and monitor appropriate restoration trajectories within this system. Recognizing and preserving this mosaic is critical in maintaining large- and small-scale peatland landscape stability and carbon balances,

particularly as long-term models suggest that the natural drying out of peatland is accelerating due to drainage (Harris et al., 2020; Leifeld et al., 2019) and climate change (Gallego-Sala and Prentice, 2013).

The approach outlined here should be readily transferable to alternative peatland settings within different parts of the global peatland climate space. Using surface motion metrics identified from the InSAR time series of peatland motion, a surface deformation space for a given peatland system can be defined. The position of ecohydrological characteristics within this

space can then be deployed to quantify the state of the peatland system and map changes with respect to climate change and management intervention. This capacity to customize the approach is valuable as it provides the means to measure peatland condition at a global scale. If realized, this would enhance our understanding of the large-scale functionality of peatland landscapes and provide the robust evidence base required for sustainable peatland management.

**Data availability**

doi: [10.17639/nott.7123](10.17639/nott.7123)

**Supplement Link**

Supplement supplied

**Author Contribution**

A.S. led the processing of the InSAR data that A.V.B. post-processed, analyzed and visualized. R.A. recorded the polygon

attributes, mapped the pools, contributed to data visualization and completed the ground surveys with C.M. D.J.L developed the overall idea of applying InSAR for this purpose. All authors were responsible for critical contributions, passing the final manuscript and editing text and figures.

**Competing Interests**

Andrew Sowter is affiliated with Terra Motion Limited. The APSIS (Advanced Pixel System using Intermittent SBAS)

method is owned by the University of Nottingham and is the subject of a UK Patent Application (No. 1709525.8) with the inventor named as Dr. Andrew Sowter; it is currently Patent Pending.



**Disclaimer**

We are not responsible for the consequences of any decisions or actions even if they have been influenced by the material and ideas in this manuscript.

**Acknowledgments**

The authors would like to thank members of the following organizations who provided access to sites for surveys or insight and local knowledge about past and present management over the study area: NatureScot Peatland ACTION, Royal Society for the Protection of Birds, Plantlife Scotland, Forestry and Land Scotland, Scottish Forestry, Welbeck Estate and Shurrery Estate. David Gee and Ahmed Athab for their assistance with the APSIS InSAR data output. "National Soil Map of

Scotland" copyright and database right The James Hutton Institute v.1_4. Used with the permission of the James Hutton Institute. All rights reserved. Any public sector information contained in these data is licensed under the Open Government License v.2.0. R.A. and C.M are funded by a Leverhulme Leadership Award (1466NS) and D.J.L, R.A. and C.M. with a NERC InSAR TOPS NE/P014100/1.

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
