# Peer review of "Identification of typical eco-hydrological behaviours using InSAR allows landscape-scale mapping of peatland condition"

_Earth Surface Dynamics, 2021_

## Author Response (AR1)

**Identification of typical eco-hydrological behaviours using InSAR allows landscape-scale mapping of peatland condition**

Bradley et al.

**Response and ACTIONS to reviewers**

**We thank the reviewers for the time and effort in reviewing this manuscript and the recognition of the significance of this work. We have considered the comments and recommendations and provided our responses in this discussion thread. Here we state our actions within the revised manuscript. Reviewer comments followed by responses and ACTIONS.**

**Reviewer #1**

This is a fascinating and useful study using novel InSAR techniques to assess blanket bog condition. The results have important implications for the application of restoration methods in peatlands.

However, the manuscript in its current form is very challenging to follow, and would benefit from a re-write of the methods section. In particular, the first few paragraphs of Section 3 (lines 284-314) should be much earlier in the paper, as they contain a useful and straightforward explanation of the metrics used. Consistency of terms and presentation throughout the manuscript would also improve readability, see detailed comments below.

The results are well written, but the discussion would benefit from more work. In particular, comparison to previous studies and an assessment of the limitations of the method would improve this section.

**RESPONSE: We agree that the paper was sometimes challenging to follow. The paper has undergone restructuring as recommended and paid attention to comments on consistency. We will discuss further our methodological limitations and refer to relevant studies where necessary.**

ACTION: Significant restructuring of the method as recommended, addition of study area section and renumbering of sections to 1. Introduction, 2. Study area, 3. Materials and Methods, 4. Results, 5 Discussion.

Method section is now 3.1 InSAR time-series preparation, 3.2 Ecohydrological classification of the sub-sites, 3.3 Mapping the state of the peatland system behaviours.

Figures have been modified, split, re-ordered and re-numbered in line with the restructuring.

(new figure > old figure).

Fig. 1 > Fig. 1a

Fig. 2 > Fig. 2

Fig. 3 > Fig. 1b-f

Fig. 4 > Fig. 3a

Fig. 5 > Fig. 3b

Fig. 6 > Fig. 3c

Fig. 7 > Fig. 3d

Fig. 8 > Fig. 4

We have added in appropriate comment into the discussion and in the results, regarding comparisons, limitations and further work; please see later responses for specific details.

I have two major concerns with this article, relating to the analysis and interpretation of results:

1. Areas of forestry are largely classified as either wet sphagnum or shrub-dominated bog (Fig. 4). This is clearly an issue with using only 3 (or 4 with irregular) categories of peatland type. The authors should definitely discuss this, and could further consider masking areas of forestry in mapping.
2. The validation of the method only tests one category, that of wet bog. At lines 380-394 the authors suggest that the method 'is converging towards 100% accuracy in identifying Sphagnum dominated pool systems in a near natural ecohydrological condition'. It must be noted, however, that a method classifying the whole study area uniformly as wet bog would give the same result using this method of validation. The authors have visually inspected the results with regard to the other categories, but the lack of quantitative validation suggests that the threshold has the potential to be set to favor the wet bog category.

**RESPONSE: Comment 1**

**The forest blocks are plantations on peat rather than on mineral soils, so we do not exclude them from the analysis. The classification under the forest is likely to be remnants of bog breathing on deep peat that is damaged in the process of tree planting, there are also open rides between the forest blocks that still support bog vegetation, and quite often there are isolated undrained pool systems within forest plantation, these can show as the different condition classes. There are generally more irregular points in forested areas as the trees do influence InSAR coherence, creating signal noise (which the MSSA can help reduce).**

ACTION: We have added detail in the text to explain this (L 363 - 374) and revised the detail and caption in Fig. 8 to show the status of the polygons.

**RESPONSE: Comment 2**

**Our aim was to present practitioners with a remote method of validation. We acknowledge that the validation is quite one sided, although we did not intend it to be that way, as explained in the text this is the one identifiable feature that will represent wet peat characteristics. For the other classes, degraded/thin and dry (stiff) peat it is much more difficult to characterise remotely using a landscape feature that will consistently relate to this class. We have considered using features such as presence of drainage channels, but these do not represent a consistent peat condition due to variations in, age, length, maintenance, and channel gradients. Similarly, with topography by considering steeper slopes >5 degrees peatland should be drier. This is problematic as these slopes naturally display variations in wetness and drainage, which is difficult to assess without a high resolution DEM. These slopes are also geographically confined to the south west of the study area creating a spatial sampling bias within the study area. So pool systems are the uniquely widespread and easily identifiable condition class and validation of more degraded classes would require fieldwork which is beyond a remote assessment and the scope of our current study. Our best solution is therefore to acknowledge and discuss this limitation. We also believe that enhancement to the detailed classified maps in Fig 4 goes some way to evidencing the potential quality of our classification.**

ACTION: We have made it clear that the current remote validation has limitations and focuses on one state of the peat and indicate what further work would be required to expand the validation (L252 -256). We highlight that (some of) the authors have specialist knowledge of the area to qualify the presence of remaining condition categories (L394-399) which also refers to detail in the figure and citations of previous work in the area.

Detailed comments:

Is the multiannual average velocity the overall/total velocity of the whole timeseries for each point? Line 298 seems to suggest that only part of the timeseries is used. It could be beneficial to add a figure showing this metric.

**L298 - It would be misleading to represent velocity on this graph (which would look like a regression) as velocity comes from the interferometry processing not the time series. It was stated in the caption that the velocity is not shown and that the source of velocity is outlined in the methods.**

ACTION: No change.

**Technical comments:**

Fig 1 – I would suggest moving figures 1b-f to later in the manuscript when the three metrics have been fully explained.

**Fig 1 – split as recommended**

ACTION: 1(b-f) is now Fig. 3.

Line 37 – The authors may wish to mention recent work on peatland hydrology using SAR backscatter, particularly work by Asmuß et al. 2019, and Lees et al. 2021.

**Line 37 – We have looked at these references and consider them to be relevant to SAR backscatter rather than SAR Interferometry techniques.**

ACTION: No Change.

Line 104 – More explanation of the point threshold could be useful.

**Line 104 – sentence now provides more detail**

ACTION: As described (L 105-107).

Line 118-121 – This PCA method could benefit from more explanation. My current understanding is that, for each time series, a 12-month moving window was applied to split the data series into multiple timeseries of length 12 months, each new time series starting with a time step of 12 days. This would give approximately 60 12-month time series over a 3-year period. The PCA analysis was then completed using this dataset – is that correct?

**Line 118-121 – Almost, the mention of PCA components is redundant and distracting information, as the reconstruction was based on the original data using the EOFs rather than PCA decorrelation. This information is not necessary and has been removed.**

ACTION: As described (L125).

Fig 2 - The amplitude of the dry bog looks larger than that of the wet bog, but this is the opposite of the explanation in Section 2.4.

**Fig 2 – There are three metrics that determine the peatland condition, high amplitude is possible in both conditions (see figure 3 plots), note that the timing and velocity also vary. There are many factors that may influence the high amplitude such as proximity to riparian zones, reed beds, peat thickness and so on, which may be specific to this time series. It would be difficult to illustrate the metrics for a subdued low amplitude time series. These examples have been chosen for clarity.**

ACTION: Editing in the text better explains that there are many combinations of values within that can occur in each state, e.g it is not just amplitude alone that defines the state of the peat (L352-361).

Line 207 – More information is needed on low, medium, and high classifications of topography.

**Line 207 – class boundaries provided**

ACTION: As described (L218).

Fig. 3 – This figure is complicated to interpret due to the mix of metrics and groupings presented in different ways. I would suggest presenting the three different metrics in a more consistent way, insofar as that is possible.

**Fig 3 – We have split the diagram up into individual parts under thematic headings and will produce the diagrams in a similar format.**

ACTION: Fig. 3a-c is now Figs. 4-7. Labels and legend text has been made consistent with the text. Fig. 4 has been revised to be more consistent with the plotting formats and expanded to show all three years.

Line 240-250 – If I have understood this correctly, 'wet' bog pixels were identified by selecting pixels with the highest amplitude and velocity, and earliest peak. Why were the field observations, sub-sites and random points, not used for this?

**Line 240 to 250 – This is not written clearly in the submission as some of the results came before the method. This is now more logically explained.**

ACTION: We now better state the choice of the reference 'soft' peat 'Within the 3-axis plot, we then chose a point with a winter peak timing, a high amplitude and extreme positive velocity, normally associated with 'soft' wet, Sphagnum peat and mapped the whole study area relative to that point. …' (L226-230). We also now discuss 'soft' and 'stiff' peat rather than just 'wet' and 'dry' (see comment 2 for reviewer 2).

Line 321 – is 'highest monthly frequency' the same as the date of annual peak (metric 1)? Sometimes months are used and sometimes date/DoY.

**Line 321 – No, this is a further aggregation of peak timing data into monthly categories.**

ACTION: Redrawing of Fig. 4 has eliminated the step of monthly aggregation and this is now redundant information excluded from the text.

Line 343 – The graphs for the first year certainly show a strong linear relationship, but I am not convinced by the other two. The authors could consider applying regression models to this data to assess the strength of relationships

**Line 343 – we will improve this diagram to show the progression in the linear relationship with respect to the levels of precipitation. We have also added a precipitation graph in the first figure illustrating the interannual precipitation levels.**

ACTION: Figure 2 now shows the regional average precipitation and the local interannual variation in precipitation to compliment the graph. Text has been improved to explain this (L330-336).

Line 33 & 409 – It might be worth including some discussion of previous attempts to measure peatland condition using remote sensing, particularly those that focus on the same area, e.g., Artz et al., 2019.

**Line 33 & 409 – We have referenced this work. However, without a thorough comparison of this optical to our InSAR derived data, which is beyond the scope of the paper, we would be speculating on the details.**

ACTION: Reference is included in the introduction and we add this point into the discussion (L410).

The authors mention early in the manuscript that the data from 2018 were excluded due to drought affecting the InSAR results. It would be good to see this explored further in the discussion, both the negative impacts of this on the reliability of the method for future use, but also the potential benefits of using the method to detect drought impacts.

**It is a fair and interesting point but this is not the intended scope of the paper. However, we agree that that it should be mentioned in the discussion section**

ACTION: we have added a comment to acknowledge this point (L428-429).

Technical comments:

Line 27 – Consistency of terms: peatlands vs peatland

**Line 27 – we use peatland**

ACTION: We default to peatland but in some cases we use peatlands where it is appropriate.

Line 97 – Define DInSAR

**Line 97 – addressed - Define DInSAR**

ACTION: Written out in full (see L100).

Line 103 and 105 - Not sure what (54) and (55) refer to?

**Line 103 and 105 – addressed, incorrect reference formats**

ACTION: As described (see L105, L110).

Line 133 – Define MSSA

**Line 133 – addressed**

ACTION: As described (see L121).

Line 192 – grammar error 'had were'

**Line 192 – addressed**

ACTION: Now edited out.

Line 246 – no comma needed after 'case with'

**Line 246 – addressed - no comma needed after 'case with'**

ACTION: As described (see L229).

Fig.3 - RAND and SS definitions need to be moved from the caption of (b) to (a).

**Fig.3 This has been addressed with the reorganisation of this figure.**

ACTION: Fig 3. now split into Fig. 4-7 and captions revised.

Line 336 – 'least negative/most negative' – unclear if this means lowest/highest values, or smallest/greatest numbers of datapoints

**Line 336 – in the method section we now state clearly that a measure of peatland swelling is a greater positive value and subsidence a greater negative value**

ACTION: As described (see L152 -154).

Line 374-378 – grammar – consider using a colon to introduce the categories, and semi-colons to separate them, rather than full stops.

**Line 374-378 – addressed**

ACTION: This part of the text has now been revised into L352 –L361.

**We thank the reviewer for the time and effort in reviewing this manuscript and the recognition of the significance of this work. We have considered the comments and recommendations and provide our responses in this discussion thread. Reviewer comments followed by responses and ACTIONS.**

**Reviewer #2**

This is an interesting study about the applicability of InSAR data for mapping and assessing peatland elevational changes and peat condition, which could provide a far more time-effective method for assessing remote peatlands worldwide. The results could also be used to enhance peatland restoration strategies.

That said, I have some reservations about the manuscript in its current form. The methods section is very difficult to follow and could benefit from some rewriting, possibly with some parts of the results section relocated there (see specific comment on L284-314 and 370-390 below). Additionally, throughout the paper, there seems to be an effort to use the technical terms for measured factors (e.g. multiannual average velocity). Given that these terms are defined (albeit quite late on in some cases), it would make it far more understandable to the reader in some cases to use the definitions in text rather than expecting the reader to constantly refer back (or in some

cases forward) to the definitions. For example, L334-335 "Where bare peat is dominant, the most negative velocities occur." Could be written as "Where bare peat was dominant, the greatest peat subsidence occurred."

**General:**

**Some of the difficulties in understanding the terminology have been addressed with the restructuring, so we now define terms earlier. We also feel that the terms should remain technical as they are relevant to the signal processing, and use of more familiar terms may cause misunderstanding with respect to the actual data being analysed and discussed. We agree with the restructuring comments and moved methods that were mixed in with the results L284-314 and L370-390**

ACTION: Significant restructuring of the method as recommended, addition of study area section and renumbering of sections to 1. Introduction, 2. Study area, 3. Materials and Methods, 4. Results, 5 Discussion.

Method section is now 3.1 InSAR time-series preparation, 3.2 Ecohydrological classification of the sub-sites, 3.3 Mapping the state of the peatland system behaviours.

Figures have been modified, split, re-ordered and re-numbered in line with the restructuring.

(new figure > old figure).

Fig. 1 > Fig. 1a

Fig. 2 > Fig. 2

Fig. 3 > Fig. 1b-f

Fig. 4 > Fig. 3a

Fig. 5 > Fig. 3b

Fig. 6 > Fig. 3c

Fig. 7 > Fig. 3d

Fig. 8 > Fig. 4

We have maintained the terminology for reasons described in the response.

I also have two concerns relating to the methodology and interpretation of results:

1. This study used four and a bit years of data (see below to comments on actual length) yet only really considered data from 3 years due to 2018 being classed as a drought year. If data had been taken over a different period (e.g. 2017 to 2021), I am not sure that 2018 would have been excluded quite so readily as May 2020 was hot and dry and April 2021 was very dry but cold. Therefore, particularly in the context of climate change and dry summers likely to become more prevalent in the UK and parts of Europe, I would like to see much greater consideration of the 2018 drought otherwise I have doubts that this method would prove particularly useful for regions where the climate is changing (and see more specific comments below).

2. Whilst I understand that pool systems will (should) always have Sphagnum and are easy to identify, I question whether assessing the predictive accuracy using just one PFT and hydrology cluster is wise (L255-264 and 380-394). Surely this is the category that is most likely to be accurate anyway, given that pools will mainly be on similar topography and slope. Why was predictive accuracy not verified by ground-truthing, since field visits are mentioned elsewhere, across a range of clusters? Also, to claim that "[Sphagnum pools]

correspond to the part of peatland systems most unequivocally associated with 'near-natural' ecohydrological condition" is possibly a dangerous assumption. Whilst this may be true for the bogs that you included in your study, it is not necessarily the case for all bogs across the world, nor even across the UK (there is quite a bit of peat formed from sedges for example). Relying on this assumption could seriously limit the applicability of your method of peatland mapping and condition assessment for global use, as you have indicated you think the method could be used for in both the abstract and discussion, and therefore I feel that further validation is necessary.

**Comment 1**

**This approach could be used to analyse climate, etc, but this is beyond the scope of this paper. We have omitted the drought year as this removes additional complications in the time series analysis in particular the potential for anomalous results if the rate of subsidence exceeds what the InSAR technique can measure. The scope of this paper is illustrating the method of characterising peatland using InSAR measures of surface motion and there is considerable potential for future research from this point.**

ACTION: We better explain the scope of the paper (L157-L159) and the choice and presentation of data (L155-165). We also acknowledge the influence of climate on the data (L429).

**Comment 2**

**The final peatland state map presents the condition of the whole peatland system, based on the physical response of the peat. We are aiming to validate the mechanical response most likely to be occurring where ideal peatland conditions are found, i.e. on soft, wet peat, which in this case is expected to be Sphagnum rich and usually associated with pools. We initially described the classes as sphagnum and shrub classes, as this is what our ecohydrological investigation clarified for the Flows, and this is what restoration experts feel comfortable to target and understand in this area. The classes could equally be ascribed appropriate plant functional groups in other (global) locations. The inclusion of vegetation in the classification description appears to give provenance to the method so we re-describe the classes as 'soft' peat, 'stiff' peat and 'modified/thin' peat states, since our classification is based on surface motion behaviour not land cover (more easily derived from optical images) thus also demonstrating an interchangeable classification for peatland globally.**

**We aimed to present a predictive accuracy of the map that could be made remotely. Pool systems are the most identifiable feature where we have most confidence in the 'wet' condition. The opposing category, dry (stiff) peat is most likely on steep slopes, but this condition is less consistent due to complexity of controlling variables, management / peat thickness / local hydrology etc. We acknowledge that the current map as presented is somewhat one sided, and now discuss the issues and outcomes of remote validation (please see also comments to reviewer 1).**

ACTION: We have made it clear that we were addressing a remote validation problem (L241), that the current remote validation has limitations (L252-256) and we suggest what would be required for the other classes (L256). We have also emphasised that we have specialist knowledge of the area, that helped us provide the interpretation of the remaining classes and map (L394-L399) and Fig. 8c.

We now use the terms 'soft' and 'stiff' peat with 'wet' and 'dry' where appropriate to help substantiate global transferability of the method.

**Specific comments:**

Figure 1 – This figure would make much more sense if b-f were given later in the paper after they had been explained. The z axis (velocity) is unclear as to which end contains negative numbers and which positive. What exactly does "frequency" (inset) mean? Is this number of obtained measurements? For the explanation of coloured dots, more clarity is needed. Why are there two different colours for front and back? Front and back of what? The graph? Also, why do these graphs only use one year of data when there is over 4 years data available?

**Figure 1 has now been split, the plots are shown after explanation of the metrics and time series. There is more explanation to clarify the remarks made above. We explain that one year is shown for clarity and we will make the axis clearer.**

ACTION: 1(b-f) is now Fig. 2, z axis redrawn. There is no reference to front and back and two colours have been removed. Frequency histograms enlarged into (a). Choice of data to display is explained (L155-L165).

L62-65 and L95-97 – these lines repeat the same information and define (some of) the same acronyms. Better to combine and only have once.

**L62-65 and L95-97: addressed**

ACTION: Repetition removed in the method restructuring.

L101-104 – whilst this may all be fully understandable and sufficient detail to someone using the software, to someone who has not, it feels as though there are lacking details/explanations. Are these default settings? If not, why were those thresholds chosen?

**L101-104: more detail and clarity added**

ACTION: As described (see L105).

L104 – presumably Wick Airport is not underlain by peat? Worth clarifying.

**L104: more detail added**

ACTION: As described (see L108).

L107-109 – this is not a full sentence and it is unclear where it is going. Consider revising.

**L107-109: sentences revised**

ACTION: As described (see L110 -113).

L118 – it says 4-year 5-month data here but the dates given in L94 do not add to this. Likewise Figure 2 legend starts 12th March 2015 to 1st July 2019 but text suggests June not July. Which is correct?

**L118: 1st July correct and made consistent in the text.**

ACTION: As described (see L98).

L118-121 – this is the only time that PCA channels are mentioned. Why are 10 calculated? Did you use them for anything else? If not, why are they mentioned?

**L118-121. We apologise for including the PCA settings. The mention of PCA components is redundant and distracting information, as the reconstruction was based on using EOFs rather than PCA channels. This information is not necessary and has been removed.**

ACTION: As described (see L125).

L122-128 – whilst I can see from Figures 2 and S1 that 2018 is different from the rest due to the drought (although not that much if just looking at the R2 line in Figure 2), I feel it would be worth giving more consideration to the results given that include 2018. Unfortunately droughts such as in 2018 are likely to become more frequent under climate change (e.g. May 2020 was very dry and warm, and then April 2021 was incredibly dry throughout much of the UK but it also happened to be very cold) and therefore if you want the model to truly represent peatland condition and apply to restoration endpoints, would drought events not need to be considered within the approach? It is also interesting that the drier bog merely showed greater amplitude in displacement rather than being completely out of pattern as the wet bog was.

**L122-L128 The purpose of our paper is to demonstrate the capacity of surface motion measured by InSAR to characterise peatland condition. Detail analysis of the response to climate and anomalous**

weather patterns would be fascinating as would be the temporal specifics of restoration but is firmly beyond the scope of this paper, we will however include this point in the discussion

ACTION: We better explain the scope of the paper (L157-L159) and the choice and presentation of data (L155-L165). We also acknowledge the influence of climate on the data (L429).

Figure 2 – it is rather confusing to have two legends with the same coloured lines in. Either it needs to be specified that one legend relates to horizontal and the other vertical lines or each line needs to be given a more distinct colour, preferably the latter since specifying lines as "grey" (L131) in the legend does not improve clarity.

**Figure 2 – Improved clarity.**

ACTION: Vertical lines kept as submission, the time series lines have now been coloured.

Figure S1 – the legend states that EOFs 1-4 and 7-8 have periodicities greater than 12 months but it looks as though 7-8 may be less. However, it is hard to tell as the quality of the graphs is poor and the axis labels need to be larger.

**Graphs redrawn more clearly as suggested and we report the periodicities as calculated by 'spectra'.**

ACTION: revised Fig. S1.

L147 – why does Figure 4 get mentioned before Figure 3? Also, the relevance of Figure 1 to the exclusion of the irregular time series needs further explanation.

**L147 – Figures and text re-ordered during restructuring and it has been made clear that the irregular class is not shown in 3d axis plots**

ACTION: Fig. 3 and caption has been revised to remove unnecessary colours. The irregular class is explained (L160-161).

L188-190 – this sounds like vegetation is separate from PFT. Please clarify.

**L188-190 - some minor edits to the paragraph reviewing the use of the word vegetation and changed to PTF where appropriate**

ACTION: Clarified with editing (L199)

Table S5 – this needs further explanation. Why is "100% - 5/5ths" by itself and seemingly different from "up to 5/5ths"?

**Table S5 – up to 5/5ths is 80% to and including 100% hits, 5/5ths is exactly 100% hits illustrating the number of buffer areas with a perfect scores. Labelling has been simplified in table to avoid confusion.**

ACTION: revised Table S5.

Figure 4 – the legend for b) states the thin/modified peat areas are hatched but they appear just to be grey (using legend from 4a). I'm a little lost as to the usefulness of 4c, particularly as forested land is the only thing that is clearly distinguishable and yet forested land was not classified on the peat condition map in 4a. Either there needs to be more explanation in the legend/text about it, or it could be removed.

**We retain the optical image for comparison as this adds context and understanding to the classification and restoration work in the area. Many of the forest to bog restoration programs have occurred since 1997, so we are observing a space time picture of recovering bog from felled forest to either 'soft' (wet) or 'stiff' (dry) bog. Some blocks appear to still be forest as the ESRI image database does not have cloud free recent images where forest has now been felled. We recognise that the image does not clearly explain that the polygons were thin peat, forest etc. The polygons have now been simplified and labelled to indicate their status at the time of the analysis e.g., forest, forest to bog, thin peat and open peatland. We are also searching for a better contemporary image to replace**

this. The forest areas are not masked out as they are on peat – please see response to Reviewer 1 for details.

ACTION: Fig. 8c retained and caption revised to add context and to illustrate the specialist knowledge we have to interpret the condition classes. Polygons in the figure have been revised and better labelled and supported with detail in the text (L363 –L374, L394 – L399).

L284-314 and L370-390 – these lines are not really results. They are mainly methods and should thus be in that section. However, they explain what you did far more clearly than the majority of the current methods section. I would suggest either beginning the methods with this and then continuing to fill in more details after or using these lines as the basis for the methods and expanding each step out from this more simple explanation. There are also some details which are given the methods section (e.g. about HCA) that are then repeated in the results. You need to decide which is which and avoid so much repetition, which should make it much clearer.

**L284-314 and L370-390 – restructuring of the paper has eliminated the repetition and issues described**

ACTION: Methods restructured (see previous comment).

L319-329 – it is very unclear as to what "monthly frequencies" refers to. Is this when the maximum peat height occurs or the number of up and down oscillations? Or something else?

**L319-329 – Monthly frequencies refers to an aggregation of maximum peat height by month not day. This aggregation into months has now been made clearer in the method section.**

ACTION: Redrawing of Figure 3a has eliminated the step of monthly aggregation.

L343-344 – I disagree. Whilst the first year appears to show strong linear relationships, the other two are far less convincing. As a minimum, I would suggest a linear regression on these data and providing $R^2$ values in the text to back up this statement.

**L343-344 – We are investigating the advantage of using a regression and different diagram formats to present our observations more clearly.**

ACTION: Graph redrawn to emphasis the relationships.

L348-350 – again, I disagree. To me, it looks like the middle of the three years shown had the greatest amplitude. Making sweeping statements across only three data points (effectively each year is a point here) is not advisable. Also, what does this mean? That some years are wetter than others?

**L348-350 – Our explanation for this has been written more clearly and we have added a precipitation graph to the manuscript as further evidence in figure 2.**

ACTION: Precipitation graph added in Fig. 2 to aid interpretation and better explanatory text added (L332 – L336).

L355-357 – this is the first time I have understood exactly where this bimodality is coming from and what it means. It might be worth explaining this earlier in the text.

**L355-357 – the document has now been restructured, which should make the bimodality and meaning more obvious. We retain the position of the sentence as this is a result.**

ACTION: These lines have now become redundant in the edits (L358 – L367).

Technical corrections:

L57 – missing comma after "(Fiaschi et al., 2019)".

**L57 – addressed**

ACTION: As described (see L57).

L97 – define DInSAR.

**L97 – addressed**

ACTION: Written out in full (see L100).

L102 and 105 – the (54) and (55) feel like they should be references but the reference list is not numbered.

**L102 and 105 – addressed**

ACTION: As described (see L105, L110).

L105 – define/give external reference to SNAPHU.

**L105 – addressed**

ACTION: As described (see L110).

L108 (and some other mentions of this paper) – why is this "L. Alshammari,"? Referencing should be consistent. Check reference software input/settings.

**L108  - addressed**

ACTION: As described (see L113).

L110 – "Each motion time series was processed as follows, to quantify the specific peatland surface motion metrics." Would read better as "Each motion time series was processed to quantify the specific peatland surface motion metrics as follows."

**L110 – addressed**

ACTION: As described (see L113).

L114 – Same for "R. Hyndman" about referencing.

**L114 – addressed**

ACTION: As described (see L117).

L117 – define MSSA.

**L117 – addressed**

ACTION: As described (see L121).

L152 – "in in" at end of line.

**L152 – addressed**

ACTION: Edited out of revision.

Section 2.2 and 2.3 appear to be a continuation of one another so would make more sense combined into one section.

**The restructuring has eliminated the heading of 2.2 of the original submission.**

ACTION: Materials and methods are now section 3.

Supplement 1.2 – this is not referred to in the main text; should it be? It also refers to Figure S3 but should to S2.

**Addressed**

ACTION: As described (see L173).

L182 – what does (64) refer to?

**L182 – addressed**

ACTION: A reference, (see L181).

Table S1 and S3 – define PFG. Also define all letter in PFG column as not all are obvious (e.g. LoA; G in brackets).

**Table S1 and S3 – addressed**

ACTION: As described (see SI doc).

Table S2 and S4 – what is the purpose of Rank Name/Name when there is highlighting to give ranks and Hydro Name (presumably final class name) as given? Needs more explanation in the legend.

**Table S2 and S4 – addressed**

ACTION: Rank name column removed as it is redundant

L188 – Typo of "category's". Should be "categories".

**L188 – addressed**

ACTION: As described (see L199).

L203 – Typo of "group's". Should be "groups".

**L203 – addressed**

ACTION: As described (see L213).

L258 – Figure 4d is not a map. Did you mean 4a?

**L258 – addressed**

ACTION: This is now referenced to Fig. 8.

Section 3.2 – there needs to be consistency in the way variables are referred to. In the first paragraph of this section, all variables are given as words. Why change to capitals for a variety of different lengths for the rest of the section, especially since most variables are one, or maximum two, words long? This makes it look very much like these parts were written by different authors who had not discussed the approach. Also, there is no need to constantly redefine abbreviations in brackets – if it is done well once, that is sufficient.

**Consistency in the text has been addressed with the re organisation of Figure 3 diagrams**

ACTION: Text and diagrams are consistently in lower case.

L321 – "September and or October". Is it "and" or "or"?

**L321 addressed - and**

ACTION: As described (see L285)

---

## Referee Report (RR1)

**Reviewer comments on esurf-2021-58 resubmission**

**General comments:**

This manuscript is much improved following the revisions and reorganisation that it has undergone – well done. In particular, the restricting of the Methods has made the whole document much clearer and I now understand why only the "soft" peat categories were virtually verified and there was some other verification of classification in the form of field visits. In some ways, it would be good to see an attempt at verification of the "stiff" peat, even if it was not as good, but I understand that choosing features from maps to do this would be difficult. I can now also see why 2018 was excluded as it has become clearer that you effectively set out to characterise a "baseline" (in essence) where major drought events would skew this aim, but I am pleased to see further consideration of this exclusion in the results and discussion.

I do still have a few comments below that I feel would improve the manuscript further, but most of them are relatively minor changes.

**Specific comments:**

Figure 1 – the map could do with more distinction between colours. Particularly, the brown is too dark and there is little difference between the dark green of forest and the slightly lighter green in the north of the map (which is what?).

Figure 2 – specifying "peak timing, t" as a "dotted line" is a little unclear given the number of dotted lines on the figure. Is the bolder horizontal dotted line used to find t or is it a? Either way, in part b, this dotted line extends outside of the May16-May17 motion year that the legend describes – why? Also, please specify if positive displacement is a rise or fall in bog surface (I assume rise but this would then mean that the bog was at its most swelled during the driest periods for stiff peat).

Figure S2 – the key to part c of the figure is very unhelpful. No reader will count 57000+ days from 1st January 1901 to try and work this out. The key needs to be changed to dates. If this is not possible in the software (which it should be), table of the dates should be overlain on the figure.

L155-164 – you say the analysis concentrated on May 2015 t May 2018 but then specify the last year of the timeseries as 2018-2019. Was 2018-2019 included or not?

L165 – if you are going to mention a 3-axis plot here, it would be useful to know where/what it is (i.e. refer to Figure X).

Table S4 – clusters 1, 3 and 4 seem to have PFGs with more than 3 names in, including categories with 0.0 in. Also, check if they match Table 2 (e.g. for Grass/rushes, G,R,S,nSp,nM when the n is not explained).

Table S5 – using percentages now makes it clearer but unfortunately the symbols chosen now mean the table makes less sense. ≤ means less than or equal to. This means that 0% is included in the ≤20% category (but is not) and everything should be included in the ≤100%. What I think you mean is that the categories should be 0%, 0.1-20%, 20.1-40%, etc. Given that the cumulative total of buffer zones column reads upsidedown, maybe reordering the table so that 100% is at the top would make more sense. Also, the 0% row should not be separate as this number is a separate category; however, it should be made clear that the 100% category is merely a subcategory of the ≤100% category otherwise the numbers do not add up.

L391-394 – this partly relates to the comments on % categories on Table S5. Whilst the explanation of the method and interpretation of this remote validation is indeed clearer than before, the way it is phrased still gives room for misinterpretation. As you were classifying the % of pixels in the 150 m buffer of a pool system that were identified as "soft" peat (L249-251), to just say that 97.9% of the pool system markers occurred within 150 m of the "soft" peat class misses quite a bit of detail. Firstly, these parts of the text need to refer to Table S5 (which is currently referred to only in the Methods). Secondly, I think it would be a lot more accurate and less open to misinterpretation/confusion to say something along the lines of "at least one pixel classed as "soft" peat was found within 150 m of 97.9% of the pool system markers".

**Technical corrections:**

L71 – comma missing after Scotland. Also not sure the location of the peatlands requires a reference.

L85 – ESRI, not ERSI.

L135 – "01 July 1$^{st}$". Need to decide on date format. Not sure why many (but not all) dates in the document have changed to an American format given this is submitted to a European journal.

L147 – the comma after "one" should be replaced by "is" to make a full sentence.

L149 – as for line 147 with "two".

L151 – remove comma after "three".

L179 – remove comma after "statistics".

L187 – remove commas outside of brackets.

L192 – "by walking across the area mapped in the polygon"? Or words to that effect – not sure the author actually walked a polygon!

L200 – PFT not PTF.

L278 – line should end in a full stop.

L299 and 302 – should "mass" be "height" or "speed"?

L318 – "reflecting **a** variable degree" missing a.

L391 – Table S5 says 97.8% but it is 97.9% here?

L437 – Repetition of "Winter" in brackets is unnecessary.

---

## Author Response (AR2)

**Return comments on esurf-2012-58 resubmission.**

We thank and appreciate the detailed attention that the reviewers have given to the resubmission and address the points as follows.

**SPECIFIC COMMENTS**

Figure 1 – Additional information has been added to the caption to improve the colour description on the image composite.

Figure 2 – Clarity improved with thicker solid lines and better separation of lines for (t) and (a). The word 'seasonal' between 'previous' and 'minimum' (now L152-3) is deleted it may imply the minimum has to be in the same motion year. Direction of movement clarified (now L133-4).

Figure S2 – Dates simplified to monthly labelling and caption altered accordingly.

L155-164 – Edited to correctly refer to the dates of the data analysed for this paper (now L165-169).

L165 – reference to Figure 3 added.

Table S4 – All classes are edited to the first three PFTs, using subscripts to differentiate similar named clusters in Table 2. All 'n' categories have been removed as it is clear from supplementary tables that there is no presence, and it also removes ambiguity with sample number $n$. Redundant text on L204 deleted accordingly (now L216).

Table S5 – Simplified and reorganised to include the 0 % and 100 % classes in the cumulative table, with class descriptions altered as suggested.

L391-394 – Text clarified with reference to Table S5 (now L399-398)

**TECHNICAL CORRECTIONS**

All suggestions acknowledged and addressed as suggested with the following specific points:

L299 and 302 – Changed 'mass' to 'height' (now L303)

L391 - Identified and corrected rounding error in table S5.